# Compositional Changes in Hydroponically Cultivated *Salicornia europaea* at Different Growth Stages

**DOI:** 10.3390/plants12132472

**Published:** 2023-06-28

**Authors:** Ariel E. Turcios, Lukas Braem, Camille Jonard, Tom Lemans, Iwona Cybulska, Jutta Papenbrock

**Affiliations:** 1Institute of Botany, Leibniz University Hannover, Herrenhäuserstr, 2, D-30419 Hannover, Germany; a.turcios@botanik.uni-hannover.de; 2Earth and Life Institute-Applied Microbiology, Unit of Bioengineering, Université Catholique de Louvain, Place Croix du Sud, 1348 Louvain-la-Neuve, Belgium; lukas.braem@uclouvain.be (L.B.);

**Keywords:** *Salicornia europaea*, compositional analysis, hydroponic cultivation, phenological stage

## Abstract

Abiotic stress conditions, such as salinity, affect plant development and productivity and threaten the sustainability of agricultural production. Salt has been proven to accumulate in soil and water over time as a result of various anthropogenic activities and climatic changes. Species of the genus *Salicornia* thrive in the most saline environments and have a wide climatic tolerance. They can be found in a variety of subtropical, oceanic, and continental environments. This study aims to establish *Salicornia europaea* as a novel source of plant-based compounds that can grow in areas unsuitable for other crops. The morphological and compositional changes in the tissues of *S. europaea* in different consecutive developmental stages have not been investigated so far. Therefore, a comprehensive study of changes during the lifecycle of *S. europaea* was carried out, following changes in the plant’s composition, including biomass yield, and soluble and insoluble compounds. For this, plants were cultivated in hydroponics for 15 weeks and harvested weekly to analyze biomass production, to determine soluble and insoluble compounds, protein content, and polyphenols. According to the results, glucan, xylan, and lignin increase with plant age, while water extractives decrease. Protein content is higher in young plants, while flavonoid content depends on the phenological stage, decreasing in the early flowering stage and then increasing as plants enter early senescence. Our results can aid in finding the optimal harvesting stage of *S. europaea,* depending on the component of interest.

## 1. Introduction

Abiotic stress conditions such as salinity and drought affect plant development and productivity and are major factors in threatening the sustainable production of agriculture. Salt has been shown to accumulate in soil and water bodies over time as a result of various anthropogenic activities and climatic changes. Soil salinity is one of the major global challenges that, together with population pressure, adverse environmental conditions, rising natural calamities, and climate change, has resulted in a reduction in usable agricultural land.

Today, the total area of salt-affected soils in the world is estimated to be around 831 million ha. Just below half of this area (397 million ha) contains high amounts of soluble salts inhibiting the growth of most major crops, while the other 434 million ha of sodic soils are associated with elevated sodium ions, poor soil structure, and low infiltration rate [1]. In both cases, an increased level of salinity in the soil, particularly of Na^+^ ions, has a series of negative effects on salt-sensitive plants (glycophytes). High salt contents in the soil reduce the water potential, making it more difficult for the plant to absorb both water and important nutrients such as K, Ca, and Fe. As a result of this, salt stress induces a water deficit in the plant, similar to the effects of physiological drought, affecting both plant growth and development [2]. The increased influx of Na in high salt conditions dissipates the membrane potential and facilitates the uptake of Cl, which is toxic for the cell. Moreover, high concentrations of Na have a series of toxic effects that cause osmotic imbalance, membrane disorganization, and the inhibition of cell expansion/division via the inhibition of some enzymes. This results in certain physiological and metabolic processes being highly impacted, including photosynthesis, respiration, transpiration, membrane properties, nutrient balance, enzymatic activity, cellular homeostasis, and hormone regulation. This leads to the production of reactive oxygen species (ROS) that cause stress conditions [3,4].

However, salt-tolerant plants (halophytes) differ in their salt tolerance and the mechanisms they use to regulate salt levels in their tissues. While most plant species are severely affected by salinity, halophytes are described as plants that can survive and complete their life cycle in environments with salt concentrations higher than 200 mM NaCl [5]. These plants constitute around 1% of the world’s flora and may tolerate salt concentrations that seriously affect 99% of the other plant species. Logically, they are naturally found in saline-sodic environments. They could replace traditional crops as potential sources of food, fuel, and fodder in salt-affected territories. Halophytes are classified as obligate, facultative, and habitat-indifferent halophytes based on their ecological characteristics. The genus *Salicornia* L., which belongs to the Amaranthaceae family, is one of the most widely distributed salt-accumulating halophytes. It consists of annual highly salt-tolerant plants found in saline environments all over the world. The genus *Salicornia* descended from the perennial *Sarcocornia* A. J. Scott during the Miocene, most likely between the Mediterranean and Central Asia, due to climate deterioration and changes in coastal locations, and started to diversify during the Late Pliocene–Early Pleistocene [6]. Because of a very complicated taxonomic diversification, significant phenotypic variation within the same species, and morphological parallelism between distinct clades, the characterization of the *Salicornia* genus is still a work in progress. Despite these issues, the number of *Salicornia* species is estimated to be between 25 and 30 [6]. *Salicornia* plants are typically small, less than 30 cm tall, and have a very simple morphology, regardless of species. *Salicornia* species occur along the European coastline from the Arctic to the Mediterranean, as well as on the Black Sea and Caspian Sea coastlines. It is also found sporadically inland where saline conditions exist. *Salicornia* spp. are found all over the world, except in Australia and South America, where other salt-tolerant Salicornioideae are prevalent [6].

The rise in the global population inevitably leads to an increased demand for food. With the high environmental cost of livestock, more and more interest is being shown in plant-based protein sources [7]. Currently, the main crops grown for protein extraction purposes include soybean and other members of the Fabaceae, wheat, corn, and other cereals. However, with the changing environment, new protein sources are highly sought after [8]. In addition to proteins, many other components found in plants could be used for food purposes or as a starting point for chemical applications [9]. Indeed, *Salicornia* spp. possess a rich composition of potential pharmaceutical molecules in addition to their protein, sugar, and healthy salt content [10].

Our work aims to establish *Salicornia* spp. as a novel source of plant-based compounds that can grow in areas not suitable for other crops. To the best of our knowledge, the morphological and compositional changes in the tissues of *S. europaea* in different consecutive developmental stages have not been investigated so far. Therefore, in this work, a comprehensive study of changes during the lifecycle of *S. europaea*, following changes in the plant composition, including biomass yield, and soluble and insoluble compounds, was carried out with the final aim of giving recommendations for the best harvesting time, dependent on the component of interest.

## 2. Results

### 2.1. Morphological Changes during Growth

The taxonomy of the *Salicornia* genus is considered one of the most challenging to determine. This is partly due to the young age of the lineages included in this taxon, but also due to the reduced weak morphological differentiation and high plasticity of the phenotypes, including morphological parallelisms between different species and variations within the same groups. With regards to *S. europaea* var. *aprica* (used in this experiment), the analysis of the external transcribed spacer (ETS) sequence shows that this ecotype effectively belongs to *S. europaea* by BLAST alignment. It has been identified as an *S. europaea* isolate with sequence identities of 99% to sequences of specimens stored in databases.

According to morphological characteristics, *Salicornia* spp. are characterized by leafless plants with green, succulent, articulated stems. The main stem and its opposing branches are composed of short, cylindrical internodes, each with a succulent, photosynthetic covering.

At the moment of transplanting in hydroponics, most of the plants have about five branches; after a week of transplanting, most of them had more than nine, and from week 2, it was practically impossible to count the number of branches. These branches continued to grow regularly, succulent and branched, until the plant entered the flowering phase. During the experiment, some plants entered the flowering stage in week 9, and by week 10, more than 50% of the plants had produced flowers (Figure 1). Sessile flowers were organized in three-flowered inflorescences per stem. The flowers were embedded in cavities and partly covered by the branches (Figure 1 and Appendix A), which are opposite.

The plants maintained accelerated growth during the vegetative stage, with succulent stems and water percentages ranging from 91% to 93% till week 9 after transplanting in hydroponics. Most of the *Salicornia* biomass was green, but the aerial biomass may turn yellowish and, in some cases, reddish due to betalain accumulation during senescence. The succulence in the tissues decreased from 88.9% in week 10 to up to 80% in week 15 (Figure 2). At the same time, the plants became more lignified and yellowish (Figure 2).

### 2.2. Biomass Yield

Fresh biomass increased rapidly from 35 g per container one week after transplanting to 2424 g per container in week 10, then, the fresh biomass remained almost constant, although dry biomass continued to increase until week 14 (Figure 3). Dry biomass ranged from 2.42 g per container after 1 week up to 452 g per container in week 14, then it remained almost constant till week 15 with a dry mass (DW) of 461 g per container. We observed that the water content of the tissues decreased as they entered the flowering stage and continued to decrease in the early senescence stage (after week 14) as the plant began to lignify.

### 2.3. Diminution of Soluble Components as Plant Age Increases

A compositional analysis was performed on *S. europaea* biomass at different ages. At every age, a clear visual difference between the water extract and ethanol extract was observed. The water extract showed an orange-brown color, while the ethanol was dark green (Figure 4). Additionally, drying of the water extract led to the formation of salt crystals, indicating that a significant part of the DW consists of water-soluble ash, in accordance with its halophytic nature. During the first weeks of *S. europaea* growth, 64.2% of the DW composition was the water-soluble fraction consisting of a mix of salt, proteins, monomeric sugars, and other water-soluble molecules. A consistent and significant reduction was observed in this fraction as the plant ages, reaching its lowest value in week 14 (33% of total DW).

After water extraction, the biomass was further extracted with pure ethanol. For this ethanol-soluble fraction, no downward correlation with plant ages was observed While it fluctuated from 12.7% of total DW at its highest to 5% at its lowest (W1 to W12, respectively), no globally consistent changes over the different ages were observed.

### 2.4. Changes in the Content of Insoluble Components with Plant Age

After extracting the water and ethanol-soluble fraction, the composition of the remaining residue was further examined after drying. Overall, an increase in this total fraction was observed with increasing plant age (Figure 5), going from 26% to 50% (W3 and W15, respectively). More precisely, it can be seen that all the examined components, except for structural ash and arabinose, contributed to this increase. 

An acid hydrolysis was performed on the insoluble residue. The levels of the monosaccharide constituents, glucan, xylan, and arabinan, of the fraction were analyzed by HPLC. Each of the measured polysaccharides followed a similar profile but were not equally abundant. For glucan and xylan, a significant increase was observed with plant age, peaking at week 15 at 11.9% and 7.8%, respectively. The arabinan levels were much lower than the other measured polysaccharides but followed a similar pattern, peaking at its highest level in week 14 at 2.97%.

Lignin, another key cell wall component, also increased with plant age. Its highest peak, however, occurred earlier at week 9 with 10.24%, dropping slightly after this point. Overall, lignin levels stayed consistently higher from week 9 onwards as compared to the first 8 weeks of growth. The acid-insoluble ash, found in the residue after Soxhlet extraction, contained different elements and constituted between 5% to 7% of the insoluble residue dry matter. No significant differences were observed between the ash content of *S. europaea* over the different weeks.

### 2.5. Protein Content Is Highest in the Youngest Plants

For the determination of the total protein content, the original ground biomass produced for the Soxhlet extractions was used. Nitrogen levels were measured using the Kjeldahl method in four replicates per plant age and multiplied by a Jones factor of 6.25. It was found that crude protein content was highest in young plants, with a peak of 22.3% at W2 (Figure 6). A continuous significant decrease in protein content was observed, stabilizing at week 6 to an average of around 12% of the total DW. A further decrease was observed at week 14, but this was not significantly different compared to the previous weeks.

### 2.6. Polyphenolic Pigment Visualization by Diphenylboric Acid-2-Aminoethyl Ester Staining

An insight into the diversity of *S. europaea* secondary metabolism was provided by a high-performance thin-layer chromatographic analysis (HPTLC). Methanol extracts were prepared from dried ground biomass and analyzed by HPTLC, with the entire experiment being repeated in triplicate. The polyphenolic profile of *S. europaea* was examined by staining with diphenylboric acid-2-aminoethyl ester (DPBA), indicating different groups of flavonoids emitting different colors (Figure 7). The green color corresponds to kaempferol derivatives (K), the orange color to quercetin derivatives (Q), the light blue to sinapate derivatives (S), and finally, the dark red indicates chlorophyll [11].

The kaempferol and the quercetin derivatives were observed in the samples from each week in all three repetitions. A similar profile was seen for the light blue sinapate derivatives. The semi-quantitative nature of the DBPA staining allows us to observe changes in the concentration of these types of compounds. A consistent decrease in staining was observed from weeks 9 to 11. In contrast, the colors appear more vibrant from week 12 onwards, indicating a potential metabolic switch in the *S. europaea* plants around this time.

### 2.7. Principal Component Analysis

According to the principal component analysis (PCA), the first component (PC1) explains 60% of the variation of the data, while the second component (PC2) explains 16.8% of the variation (Figure 8). Together this means that 76.8% of the variation of the data is explained by both components, which is relatively high. It can be observed that residues, biomass, insoluble components, lignin, and sugars are positively related to plant age, while protein content, water content in plant material, as well as water extractives, are negatively correlated with plant age.

According to the correlation analysis, lignin is highly correlated with residues (0.81), xylose (0.71), and glucose (0.76), and these increase with plant age (Appendix A). Xylose and glucose are highly correlated (0.97) with each other. Protein and lignin are negatively correlated (−0.76) as a result of a higher protein concentration in the early stages of the plants and a higher lignin content in mature plants. This section may be divided by subheadings. It should provide a concise and precise description of the experimental results, their interpretation, as well as the experimental conclusions that can be drawn.

## 3. Discussion

### 3.1. Main Changes in Plant Growth and Morphology

*Salicornia europaea* plants were cultivated for 15 weeks after transfer to hydroponic cultures, and aboveground biomass was harvested each week. The different developmental stages can be described as being undertaken for several established crop species [12]. The plants were transferred to the hydroponic culture at stage 2 (formation of side shoots). In the conditions of this experiment, the plants reached stage 4 (development of harvestable vegetative plant parts) about 9 weeks after transplanting. At week 9 inflorescence emerged (stage 5); at week 10, the plants were flowering (stage 6). Until week 10, a constant growth rate was observed. After week 10, plants continued to grow but at a slower rate as they began to enter a maturation stage indicated by flowering. The plants began to turn yellowish, probably due to a decrease in chlorophyll. Chlorophyll breakdown is an important catabolic process of plant senescence [13]. Lim et al. [14] reported that the loss of green color visually marks the initiation of metabolic changes that occur during senescence. This is in agreement with the observations in this study. After this stage, the amount of water in the plants decreases drastically. This process is accompanied by a decreasing amount in the extractives fraction and an increase in the structural polysaccharides (glucans and xylans) and lignin. Lignin and its related metabolism play important roles in the growth and development of plants. As a complex phenolic polymer, lignin enhances plant cell wall rigidity and hydrophobic properties and promotes mineral transport through the plant’s vascular bundles [15]. Lignin deposition depends on the cell type, the developmental stage, and the species [16]. Our findings are in line with Cybulska et al. [17], where similar lignin amounts ranging from 7.44 g/100 g TS to 23.63 g/100 g TS were found in *S. bigelovii,* depending on the treatment and fraction analyzed. During weeks 11 till 13, the development of fruits and the ripening of seeds were observed, followed by senescence and the beginning of dormancy [12].

### 3.2. Changes in Protein Content

The protein content was higher in the youngest plants, which is probably due to the active primary growth leading to more cell divisions and, therefore, more cytoplasmic content. In addition, senescence is accompanied by a decline in photosynthesis and the massive degradation of cellular proteins [18]. A sharp decline in the content of chlorophylls, carotenoids, and proteins in the shoot was also noticed in another study in *Salicornia brachiata* [19] in the middle and late stages of senescence in comparison with the early stages. The degradation of chlorophylls and breakdown of saccharides, lipids, and proteins may be increased due to the activation of hydrolytic enzymes during senescence, whereas photosynthesis as well as protein synthesis decrease [20,21]. The decline of the protein content may be attributed to the increased proteolysis [22], and the decline in the nitrogen content of the senescing shoots is also related to seed filling as part of monomers resulting from the breakdown processes are used to synthesize storage polymers in the seeds. The seeds of *S. europaea*, for example, contain high amounts of lipids [23].

### 3.3. Changes in Secondary Metabolites

According to the results of polyphenolic pigment by diphenylboric acid-2-aminoethyl ester staining, it is observed that there was a decrease in the metabolites, including kaempferol, quercetin, and sinapate derivatives from week 9 to 11, then an increase was observed from week 12 onwards, which was related to the early senescence stage. The decrease in metabolites from week 9 to 11 may be related to the beginning of the flowering stage. So far, there are no studies in *S. europaea* on this issue, but Chepel et al. [24] investigated the changes in the content of phenolic compounds in *Calluna vulgaris* and reported that at the flowering stage, a decrease in flavonoid content by about a third was observed in all studied organs. This indicates an expense of these compounds on the physiological and adaptive processes. Biosynthesis and accumulation of secondary metabolites in plants, including phenolic compounds, depend on a number of factors, such as the growth stage and growing conditions. Therefore, their content in the plant tissues changes in the course of the plant’s growth and development. This decrease in the content of flavonoids at the beginning of the flowering stage can be associated not only with the slowdown in biosynthesis but also with the translocation of some metabolites from the succulent stems to specialized organs like flowers; however, more research is needed to corroborate this. In addition, some monomeric phenols serve as the initial components for lignin synthesis, being this an important component of the secondary cell wall.

There is no consensus on which specific week is best to harvest, based on just the concentration of the different organic fractions. For the total yield of each organic compound, the biomass production per plant or per unit area should be taken into account. For example, the concentration of protein decreases with the age of plants; however, the plant biomass increases. Therefore, the total yield of protein increased from 9.50 g/m^2^ in week 2 to 393.69 g/m^2^ in week 14; then, it remains almost constant till week 15 with a value of 395.30 g/m^2^. In the case of sugars, the maximum yield is reached between weeks 14 and 15 after transplanting (Figure 9). If the interest is only fresh biomass production, the optimal harvest time would be week 10 under the conditions described above, with a yield of 2424.26 g fresh mass per container (20.20 kg/m^2^).

Therefore, if the interest is to produce a greater amount of these compounds, it would be recommended to harvest the plants between weeks 14 and 15. However, it should be taken into account that if the interest is on other primary/secondary metabolites, the harvest age of the plants could be very different. In addition, metabolite concentrations can also be manipulated in *S. europaea* plants, increasing their concentration under high salinity conditions, as reported by Boestfleisch et al. [25]. Hulkko et al. [26] also reported that the biomass and the amounts of organic compounds like lignin, protein, carbohydrates, lipids, and organic acids can vary in *S. europaea* as a function of salt concentration. From a technical–economic point of view, the grower would have to calculate the cultivation area to produce the desired yield and the cost of production that each additional week represents, which will depend on local conditions. Finally, in the case of secondary metabolites, the concentration changes depending on the growth stage (such as flowering or a stress event like that reported above), so if this plant is to be used in a biorefinery-type approach, the focus should be on the desired metabolites.

## 4. Materials and Methods

### 4.1. Plant Cultivation

The agronomic handling from sowing through transplanting in hydroponics was carried out as described by Buhmann et al. [27]. On 14 July 2021, *Salicornia europaea* var. *aprica* was transplanted in hydroponic systems at the Institute of Botany (52°23′42″ N; 9°42′13″ E), Leibniz University Hannover (LUH), Germany, with a salt concentration of 15 g/L NaCl in the nutrient solution. The salinity of 15 g/L NaCl was chosen based on previous results, where this salt concentration has been shown to be optimal for *Salicornia europaea* [26]. Each experimental unit consisted of eight plants per container. Polypropylene containers (L400 × W300 × H175 mm) with a capacity of 16 L were used. Each container had 13 L of modified Hoagland solution, containing 606 mg/L KNO_3_, 944 mg/L Ca(NO_3_)_2_·4H_2_O, 230 mg/L NH_4_H_2_PO_4_, 246 mg/L MgSO_4_·7H_2_O, 3.73 mg/L KCl, 1.55 mg/L H_3_BO_3_, 0.34 mg/L MnSO_4_·H_2_O, 0.58 mg/L ZnSO_4_·7H_2_O, 0.12 mg/L CuSO_4_·5H_2_O, 0.12 mg/L MoNa_2_O_4_·2H_2_O, and 9.16 mg/L Fe-EDDHA (0.56 mg Fe/L). The water was aerated constantly by small compressors and one air stone in the middle of each tank (Eheim, Deizisau, Germany). The hypocotyl was fixed with soft foam in 35 mm holes. The water level was adjusted constantly in each tank with tap water to compensate for the evapotranspiration. The halophytes were grown in a greenhouse at temperatures varying between 21 °C and 35 °C during the night and day, respectively. A total of 14 h of artificial light was provided (sodium vapor lamps, SON-T Agro 400W, Philips, Amsterdam, The Netherlands). Light intensity ranged from 110 µmol/m^2^ s to 630 µmol/m^2^ s depending on the time of the day and the weather conditions.

### 4.2. Plant Harvest and Biomass Determination

During the first 11 weeks in hydroponics, plants from three containers were harvested each week and then one container per week. After harvesting, the aboveground fresh biomass was determined, and the plant material was dried at 60 °C for 48 h prior to determining the dry biomass (DW) and composition analysis. The water content in the plant tissue was calculated as the following: water content(%) = (FM − DW)/FM × 100, where FM is the fresh mass and DW is the dry mass.

### 4.3. Compositional Analysis

Experimental protocols are based on the American Society for Testing and Materials (ASTM) and National Renewable Energy Laboratory (NREL) standard protocols. The compositional analysis was started by extracting 5 g of dried biomass with 150 mL of water in the Soxhlet apparatus for 8 h. An additional extraction was performed on the residue in the thimble using 150 mL of ethanol for 8 h in the Soxhlet apparatus. The solvents were evaporated in a rotary evaporator (Heidolph, Schwabach, Germany), and the yields were then measured by an accurate mass balance. Dry matter and ash contents were determined by drying biomass while ashing was performed at 550 °C. Quantification of structural carbohydrates and lignin was performed by the addition of 72% sulfuric acid to the solvent-free biomass and subsequently hydrolyzation by autoclaving. Their cellulose, hemicellulose, and lignin composition were determined according to the NREL protocol for the determination of structural carbohydrates and lignin in biomass [28]. The sugar concentration was measured by HPLC (Shimadzu, Kyoto, Japan) equipped with a column Aminex HPX-87H (BioRad, Hercules, CA, USA) running an isocratic flow of aqueous H_2_SO_4_ (5 mmol/L) at 0.7 mL/min for 30 min. Detection was achieved by a refractive index detector set at 55 °C. The dry weight (DW) of the biomass sample was calculated using the average total solid (TS) content and the recorded biomass weight introduced in the thimble.

### 4.4. Determination of Nitrogen Content

To determine total nitrogen (%), the Kjeldahl method was used. In short, 2 mL (if liquid) or around 0.4 g (if solid) of the sample was mixed with 10 mL of sulfuric acid (95–99% Sigma Aldrich, Taufkirchen, Germany) in the presence of a 5 g Wieninger catalyst tablet (Sigma Aldrich). Subsequently, the acid was heated to 420 °C for 40 min in a KJELDATHERM Manual from Gerhardt. Distillation of the digested samples was performed on a VAPODEST machine from Gerhardt, using a 2% boric acid solution to capture the liberated ammonia. Finally, titration was performed using a 0.5 M Hydrochloric acid solution, and total nitrogen% was calculated according to $Nitrogen\left(\%\right)=\frac{\left(V_{1}-{\bar{V}}_{0}\right)\times N\times 1.4007\;}{m}$, and then multiplied by a standard Jones factor of 6.25 to calculate the protein concentration (%) [29]. 

### 4.5. Statistical Analysis

All statistical analyses were conducted using R, version 3.1.1 (R Core Team, Vienna, Austria) and InfoStat software, version 2020e (InfoStat Team, Cordoba, Argentina). The effects of the plant age on the different parameters were analyzed through a one-way analysis of variance (ANOVA). The Tukey multiple comparison test, with a significance level of = 0.05, was carried out to determine which means differ from the rest. Multivariate data analysis was undertaken in order to observe trends, jumps, and outliers and express the information as a set of summary indices (principal components). The first two principal components were used in order to plot the data in two dimensions and to visually identify clusters of closely related data points. The Pearson coefficient was used to evaluate the correlation between the different variables.

## 5. Conclusions

The biochemical composition and morphology of *S. europaea* depended on the phenological stage. Tissue water content decreased with age and lignification increased with the plant age, entering the flowering stage from week 9 under hydroponic conditions. Fresh biomass production increased rapidly from transplanting until week 10 with a production of 2424 g per container (303 g/plant), then production stabilized, but the highest dry matter production was obtained in week 15 with a yield of 461 g per container (58 g/plant).

Glucan, xylan, and lignin content increased with plant age. Flavonoids decreased from week 9 to 11, which was the stage where the plants entered flowering, increasing again from week 12 onwards. Protein concentration decreases with plant age, but biomass production per plant or per unit area should be taken into account. For example, the concentration of protein decreased with the age of plants; however, the plant biomass increased. In the case of sugars, the maximum yield is reached between weeks 14 and 15 after transplanting. Based on the results, *S. europaea* can be cultivated under saline conditions, and the biomass can be used for different purposes, being a source of valuable secondary metabolites; however, the changes in concentration depend on the growth stage.

## Figures and Tables

**Figure 1 plants-12-02472-f001:**
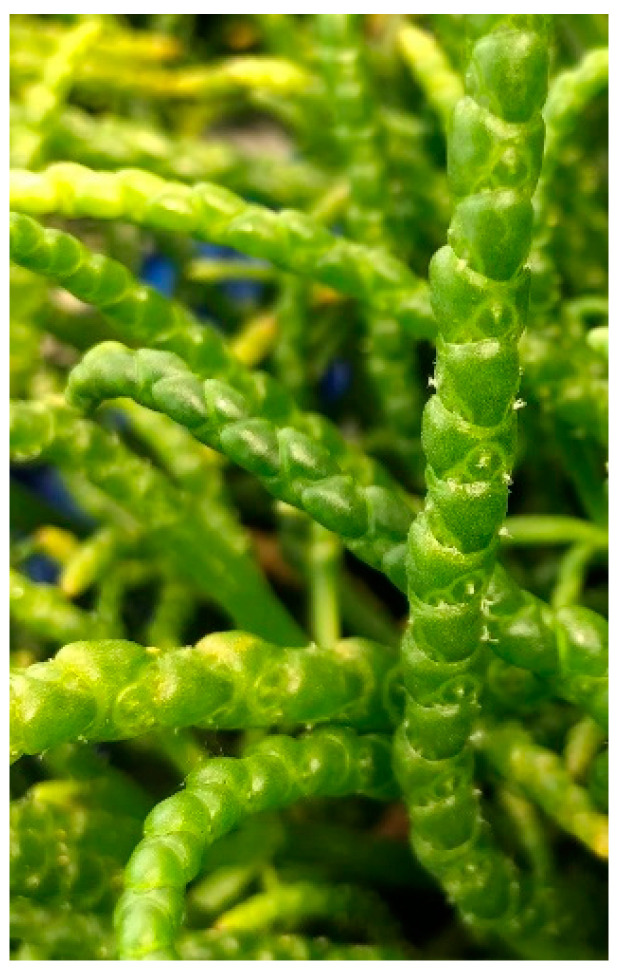
Flowering *Salicornia europaea* plant, week 10 after transplanting in hydroponics.

**Figure 2 plants-12-02472-f002:**
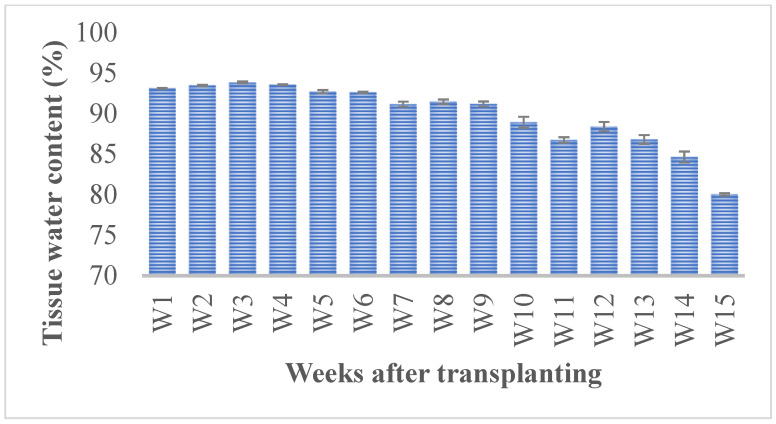
Water content (%) in *Salicornia europaea* tissues. Bars represent the average of three replicates ± standard error.

**Figure 3 plants-12-02472-f003:**
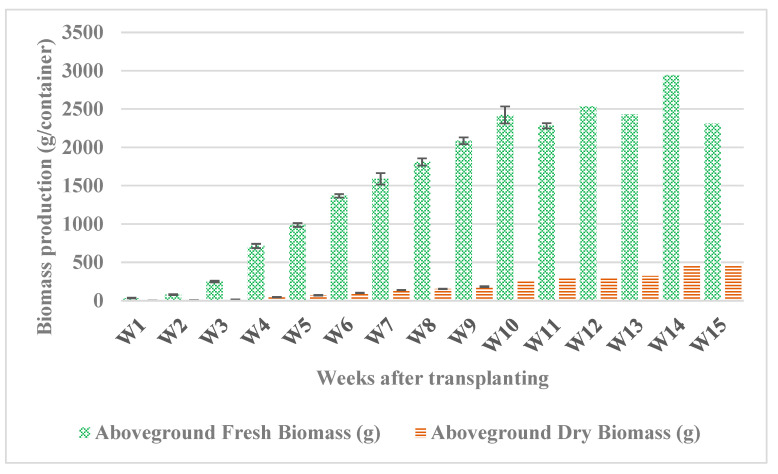
Fresh and dry biomass production of *Salicornia europaea* under hydroponic conditions for 15 weeks. Bars from week 1 (W1) till week 10 (W10) represent an average of three replicates (containers) ± standard error. Bars from W12 to W15 represent one replicate.

**Figure 4 plants-12-02472-f004:**
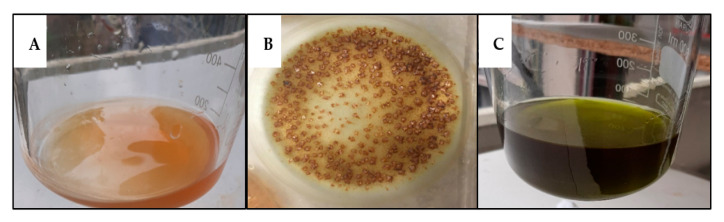
Soxhlet extracts of *Salicornia europaea*. (**A**). Water Soxhlet extract from dried biomass. (**B**). Salt crystals appear in the dried water extract. (**C**). Ethanol Soxhlet extract shows a clear green color.

**Figure 5 plants-12-02472-f005:**
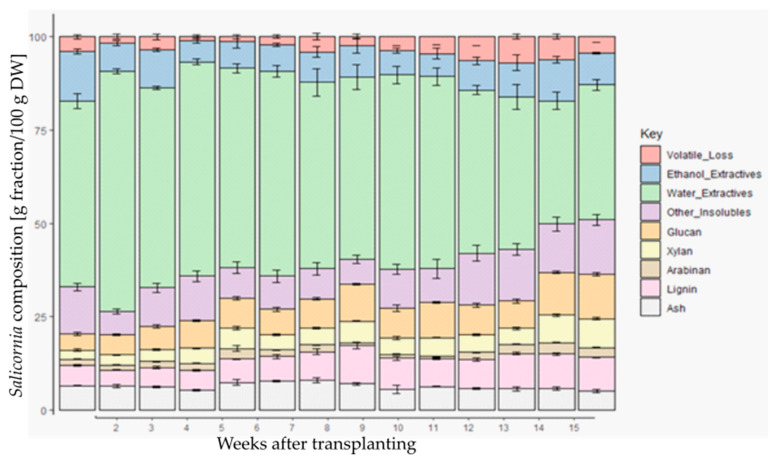
Composition of *Salicornia europaea* at different points of its development. Results are shown as gram fraction per 100 g dry weight (DW). Statistical significance was checked with ANOVA analysis with Post hoc Tukey, a summary of which can be found in the annex.

**Figure 6 plants-12-02472-f006:**
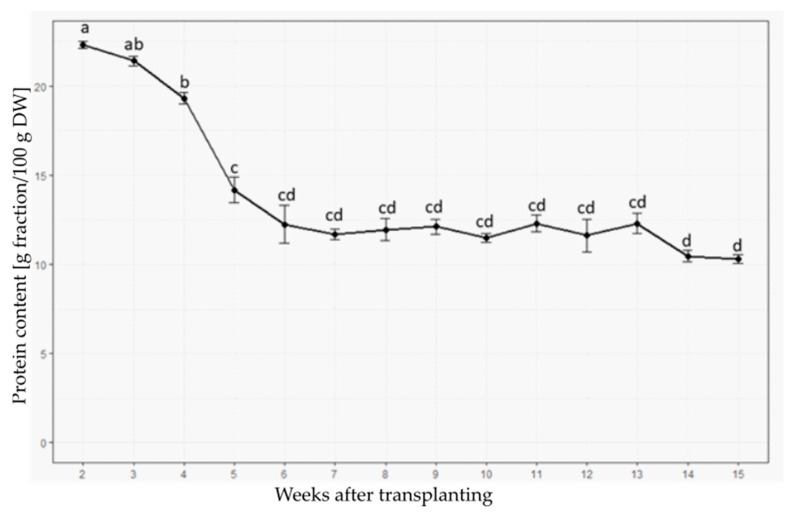
Overview of protein content as gram fraction per 100 g DW at different plant ages. Protein content was estimated by Kjeldahl nitrogen measurement and multiplication with a Jones factor of 6.25. Values with common lower-case letters are not significantly different (*p* > 0.05).

**Figure 7 plants-12-02472-f007:**
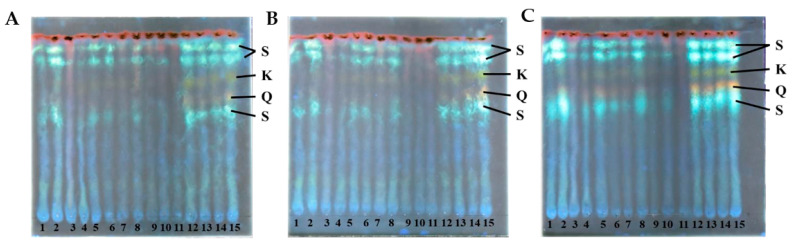
Profiles of polyphenolic pigments determined by high-performance thin-layer chromatography (HPTLC)—staining with diphenylboric acid-2-aminoethyl ester (DPBA). (**A**–**C**) correspond to repetitions of the experiments, while the numbers (1–15) represent the samples from W1 to W15 (1–15). K = kaempferol derivatives, Q = quercetin derivatives, S = sinapate derivatives.

**Figure 8 plants-12-02472-f008:**
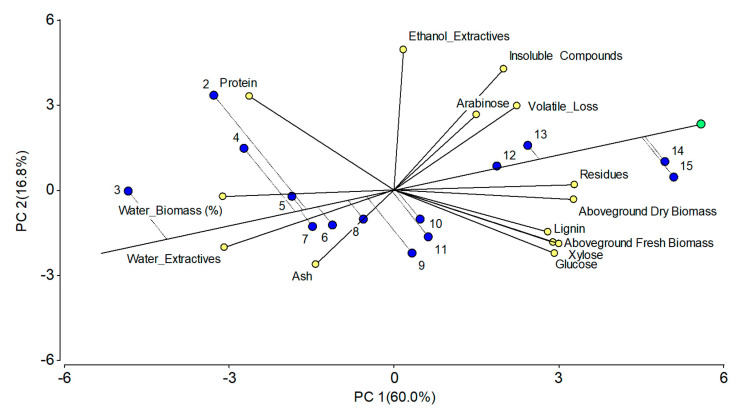
Principal component analysis for the compositional parameters in *Salicornia europaea* at different growth stages. Blue dots indicate the different harvesting points (from 2 till 15 weeks after transplanting).

**Figure 9 plants-12-02472-f009:**
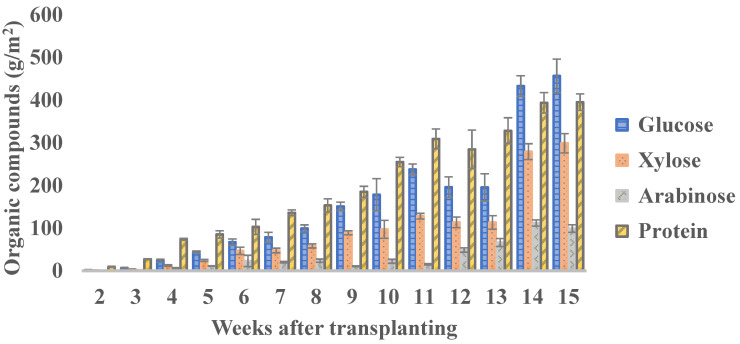
The yield of protein and sugars in different harvesting points of *Salicornia europaea.* Bars represent the mean value of four replicates ± SD.

## Data Availability

The original data sets are available from the authors.

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
