# Peer review of "Compositional Changes in Hydroponically Cultivated Salicornia europaea at Different Growth Stages"

_plants, 2023, doi:10.3390/plants12132472_

Round 1
Reviewer 1 Report
In this manuscript, the compositional components in the plant extract of Salicornia europaea were measured, which was found to be varied at different growth stages on the lifecycle of the plant. The authors therefore suggested that for extracting different compositional product, the plant harvest time should be different. However, after reading the paper, I have some concerns that should be addressed:
1. Although the “morphological changes” was highlighted in the title, the paper did not show a rigourous result on morphological changes of S. europaea. In figure 1, the photos are aimless so that reader cannot obtain any useful information. It would be grateful that all photos are taken in a same direction while the background are same and clear. The details such as inflorescence could be magnified and presented in other ways.
2. In the part of Material and Methods: the method of calculating water content is missing.
3. An explanation on the difference between oven dried weight (ODW) and “dry mass (DW)” presented in line 145 is needed.
4. In figure 7, the figure number is not presented in a correct way.
5. In figure 8, the spot represented sample of week 1 is missing.
6. In table S1, the decimal point should be “.”.
7. Some conclusions in the paper are lack of academic rigor. For example, the estamated yield of 20.20 kg/m2 was deduced from the fresh weight under the planting density of eight plants per container, which may be different when the planting density changes. The unit of “kg/plant” is suggested to evaluate the biomass of the plant in this work.
8. The format of references should be checked carefully.
The language should be polished by a native English speaker or a professional English editing agent.
Author Response
Dear Editor, dear reviewers,
Thank you for carefully reading the manuscript. We adressed each point. You find our responses to each point. I hope the manuscript is now acceptable for publicatoin in Plants.
Reviewer 1
In this manuscript, the compositional components in the plant extract of Salicornia europaea were measured, which was found to be varied at different growth stages on the lifecycle of the plant. The authors therefore suggested that for extracting different compositional product, the plant harvest time should be different. However, after reading the paper, I have some concerns that should be addressed:
- Although the “morphological changes” was highlighted in the title, the paper did not show a rigourous result on morphological changes of S. europaea. In figure 1, the photos are aimless so that reader cannot obtain any useful information. It would be grateful that all photos are taken in a same direction while the background are same and clear. The details such as inflorescence could be magnified and presented in other ways.
R: Salicornia is a species that does not behave in a similar way to other conventional plant species, it does not present clearly differentiated morphological changes, so it is difficult to differentiate morphologically in each stage. Therefore, we modified the title to “Compositional changes in hydroponically cultivated Salicornia europaea at different growth stages”. Unfortunately, the photos were not taken in the same way every week. The reviewer's recommendation has been partially followed, where in figure 1 the magnified photograph of the Salicornia inflorescence has been left, and the previous figure where the 15 photographs are, has been moved to supplementary material.
- In the part of Material and Methods: the method of calculating water content is missing.
R: It was added in the new paragraph “4.2. Plant harvest and biomass determination”
- An explanation on the difference between oven dried weight (ODW) and “dry mass (DW)” presented in line 145 is needed.
R: It is the same, fresh plants were dried for 2 days in an oven at 60°C until constant weight. To avoid confusion, we removed the ODW term and just describe it as dry weight (DW).
- In figure 7, the figure number is not presented in a correct way.
R: the lettering (A,B,C) were placed on top of the image.
- In figure 8, the spot represented sample of week 1 is missing.
R: week 1 was not added as we lacked enough biomass to do all the compositional analysis for this week. So the description of the image was changed to “…different harvesting points (from 2 till 15 weeks after transplanting)”
- In table S1, the decimal point should be “.”.
R: Done
- Some conclusions in the paper are lack of academic rigor. For example, the estamated yield of 20.20 kg/m2 was deduced from the fresh weight under the planting density of eight plants per container, which may be different when the planting density changes. The unit of “kg/plant” is suggested to evaluate the biomass of the plant in this work.
R: The biomass yield was changed to g/plant in the conclusion section
- The format of references should be checked carefully.
R: The format was followed according to the journal
Comment on the English language (in the formulars): Should be corrected by a native speaker.
R: The lanuage was checked by a native speaker.
Reviewer 2 Report
Clear and well-structured manuscript with specific, well-defined objectives and appropriate methodology. In any case, the authors should justify why they have grown Salicornia in hydroponics and why they have chosen a salinity of 15g per liter of NaCl. Apparently, the culture medium was not renewed during the entire experience, and therefore we do not know if any nutrient could be limiting at the end of the experience and may have had some influence on the results obtained. Nothing is said about the belowground biomass either. The evolution of this belowground biomass and its chemical composition may also be of interest.
Notes:
-Line 64: Change “The Salicornia genus” for “The genus Salicornia L.”
-Line 67: Change “Sarcocornia” for “Sarcocornia A.J. Scott”
-Line 108: “Salicornia spp. are characterized…”
-Lines 141-142: “Fresh biomass increased rapidly from 35 g per container in week 1 after transplanting to 1424 g per container in week 10”. However, the graph in Fig. 3 indicates an average of 2500 g per container that week, check! This same figure should explain why from weeks 12 to 15 only one replica was used...
-Line 177: After extracting the water and ethanol-soluble fraction the composition of the remaining residue was further examined after drying the oven-dried weight (ODW).
-In Figure 8 where is the blue dot corresponding to the first week?
-Line 103; 360: “aprica” in italics.
Author Response
Dear Editor, dear reviewers,
Thank you for carefully reading the manuscript. We adressed each point. You find our responses to each point. I hope the manuscript is now acceptable for publicatoin in Plants.
Reviewer 2
Clear and well-structured manuscript with specific, well-defined objectives and appropriate methodology. In any case, the authors should justify why they have grown Salicornia in hydroponics and why they have chosen a salinity of 15g per liter of NaCl. Apparently, the culture medium was not renewed during the entire experience, and therefore we do not know if any nutrient could be limiting at the end of the experience and may have had some influence on the results obtained. Nothing is said about the belowground biomass either. The evolution of this belowground biomass and its chemical composition may also be of interest.
R: It was grown under hydroponic conditions mainly for two reasons: 1. to better control the salinity of the culture medium, which would be much more difficult using soil. 2. Salicornia cultivation can be used in conjunction with fish cultivation (aquaponics), or to treat saline water so there is some potential for hydroponic cultivation. The salinity of 15 g/L was chosen based on previous results, where a salinity of 15 g/L has been shown to be optimal for Salicornia. This was added in paragraph 4.1 plant cultivation: “The salinity of 15 g/L NaCl was chosen based on previous results, where this salt concentration has been shown to be optimal for Salicornia europaea [26].”
Although the roots may be interesting, they were not within our focus of interest. In the future, investigations will be carried out taking these important recommendations into account.
Notes:
-Line 64: Change “The Salicornia genus” for “The genus Salicornia L.”
R: Corrected
-Line 67: Change “Sarcocornia” for “Sarcocornia A.J. Scott”
R: Corrected
-Line 108: “Salicornia spp. are characterized…”
R: Corrected
-Lines 141-142: “Fresh biomass increased rapidly from 35 g per container in week 1 after transplanting to 1424 g per container in week 10”. However, the graph in Fig. 3 indicates an average of 2500 g per container that week, check! This same figure should explain why from weeks 12 to 15 only one replica was used...
R: It was a typo error, the correct value is 2424, and it has already been corrected. About the one replica it was due to the lack of enough material and space, so for these weeks only one replicate was available but with 8 plants each, however we consider this is not important to explain in the figure.
-Line 177: After extracting the water and ethanol-soluble fraction the composition of the remaining residue was further examined after drying the oven-dried weight (ODW).
R: Corrected see reply to point 3 for reviewer 1.
-In Figure 8 where is the blue dot corresponding to the first week?
R: Added
-Line 103; 360: “aprica” in italics.
R: Corrected
Comment on the English language (in the formulars): Should be corrected by a native speaker.
R: The lanuage was checked by a native speaker.